# Time Evolution Features of Entropy Generation Rate in Turbulent Rayleigh-Bénard Convection with Mixed Insulating and Conducting Boundary Conditions

**DOI:** 10.3390/e22060672

**Published:** 2020-06-17

**Authors:** Yikun Wei, Pingping Shen, Zhengdao Wang, Hong Liang, Yuehong Qian

**Affiliations:** 1Joint Engineering Lab of Fluid Transmission System Technology, Faculty of Mechanical Engineering and Automation, Zhejiang Sci-Tech University, Hangzhou 310018, China; yikunwei@zstu.edu.cn (Y.W.); shenpp@cjlu.edu.cn (P.S.); 2Department of Physics, Hangzhou Dianzi University, Hangzhou 310018, China; 3School of Mathematical Science, Soochow University, Suzhou 215006, China; yuehongqian@suda.edu.cn

**Keywords:** entropy generation rate, thermal plume, mixed boundary conditions, heat transfer

## Abstract

Time evolution features of kinetic and thermal entropy generation rates in turbulent Rayleigh-Bénard (RB) convection with mixed insulating and conducting boundary conditions at *Ra* = 10^9^ are numerically investigated using the lattice Boltzmann method. The state of flow gradually develops from laminar flow to full turbulent thermal convection motion, and further evolves from full turbulent thermal convection to dissipation flow in the process of turbulent energy transfer. It was seen that the viscous, thermal, and total entropy generation rates gradually increase in wide range of *t*/τ < 32 with temporal evolution. However, the viscous, thermal, and total entropy generation rates evidently decrease at time *t*/τ = 64 compared to that of early time. The probability density function distributions, spatial-temporal features of the viscous, thermal, and total entropy generation rates in the closed system provide significant physical insight into the process of the energy injection, the kinetic energy, the kinetic energy transfer, the thermal energy transfer, the viscous dissipated flow and thermal dissipation.

## 1. Introduction

The Rayleigh–Bénard (RB) convection is one of most classical natural convections [1,2,3,4,5], which widely occur in a range of natural and industrial applications [1,2], such as in the Earth’s core and mantle, atmosphere, oceans and stars, nuclear reactors, crystallization processes, solar heating devices and so on. The RB convection has been extensively investigated by several experiments in the last few decades [6,7], mostly in slender cells of aspect ratio smaller than or equal to unity in order to reach the largest possible Rayleigh numbers (*Ra*) or to reveal the detailed characteristic mechanisms of turbulent viscous dynamic and heat transport near the walls or central domains [3,4]. The detailed dynamical and statistical insights of the included turbulent transport and their coherent structures also have been increasingly studied in detail by direct numerical simulations (DNS) [8,9].

It is well known that when the DNS of involving no parametrization of subgrid-scale is carried out, all the dynamically important scales are resolved to faithfully represent the flow. Bailon et al. [10] and Schell et al. [11] reported the derived resolution criteria, thus starting the pioneering work [10,11]; subsequent refinements of this criterion were studied [12,13] and the fine boundary layer dynamics resolution were the main focus [14]. It is only recently that the focus of DNS investigations was presented by the bulk of the research in a convection cell with detailed discussions of the scaling statistics and properties of the dissipation fields [15].

In order to understand the global flow and heat transport loss mechanisms in detail, we will present the detailed statistical characteristic mechanisms of the velocity and temperature gradient fields, in particular the related viscous and thermal local entropy generation rates. The viscous and thermal local entropy generation rates as a criterion are used to provide insight into the local viscous and thermal flow loss in the flow field [16,17,18,19,20,21,22,23,24,25,26,27,28,29,30]. The viscous and thermal components of the local entropy generation rate are derived in the two-dimensional Cartesian space [30]. Their expressions are as follows, respectively [20,28].
(1)$$\overset{\cdot }{S_{u}}=\frac{\mu }{\theta }\left\{2\left[{\left(\frac{\partial u}{\partial x}\right)}^{2}+{\left(\frac{\partial v}{\partial y}\right)}^{2}\right]+{\left(\frac{\partial u}{\partial y}+\frac{\partial v}{\partial x}\right)}^{2}\right\}$$
(2)$$\dot{S_{\theta }}=\frac{k}{\theta^{2}}\left[{\left(\frac{\partial \theta }{\partial x}\right)}^{2}+{\left(\frac{\partial \theta }{\partial y}\right)}^{2}\right]$$
where μ is the dynamics viscosity of fluid, k denotes the thermal conductivity, θ is the temperature, u represents the *x*-direction velocity and v is the *y*-direction velocity, respectively. The total entropy generation rate is the summation of the viscous and thermal entropy generation rates, its expression is as follows [22]:(3)$$\dot{S}=\dot{S_{u}}+\dot{S_{\theta }}$$

The Bejan number (*Be*) is regarded as an effective approach to judge the importance of heat transfer irreversibility in the domain [23]. Rejane et al. proposed the contribution of heat transfer entropy generation on over all entropy generation by using the Be [27]. The Be is defined by the following equation [29]:(4)$$Be=\frac{\dot{S_{\theta }}}{\dot{S}}$$

The range of *Be* is from 0 to 1. When Be is equal to 0, the irreversibility is dominated by fluid friction. Correspondingly, the irreversibility is dominated by heat transfer when the Be is equal to 1. The irreversibility due to heat transfer dominates in the flow when the Be is greater than 1/2. Correspondingly, *Be* < 1/2 implies that the irreversibilities due to the viscous effects dominate the processes. Meanwhile, it is also noted that the heat transfer and fluid friction entropy generation are equal in *Be* = 0.5 [27].

A wide variety of thermal plumes caused by the buoyancy in turbulent RB convection with mixed insulating and conducting boundary conditions play dominant role in the heat transfer. Once time evolution of the heat transfer has been described and understood in classical turbulent RB convection, the time evolution features of kinetic and thermal entropy generation rates in turbulent RB convection with mixed insulating and conducting boundary conditions still be further expanded. How does the mixed insulating and conducting boundary conditions reduce or improve the time evolution features of entropy generation rate? The mixed insulating and conducting boundary conditions considerably affect the time evolution characteristics of thermal plumes and entropy generation rate.

Based on above discussions, our work mainly focuses on the effect of the mixed insulating and conducting boundary conditions on the time evolution features of thermal plumes, the viscous, thermal and total entropy generation rates. The physical insight features of kinetic and thermal entropy generation rates with time evolution are discussed in in turbulent RB convection with the mixed insulating and conducting boundary conditions, which mainly tried to understand the dynamics of fluid. The remainder of this paper is divided into the following parts. In Section 2, the thermal fluid dynamics equations and numerical method will be briefly depicted. In Section 3, the detailed results of numerical simulation and some discussions are presented. Finally, some conclusions are addressed.

## 2. Convection Diffusion Equation and Numerical Method

In this section, the convection diffusion equation of thermal fluid, and numerical method of solving them are represented, respectively.

### 2.1. Convection Diffusion Equation of Thermal Fluid

The convection diffusion equation of thermal fluid is the classical Oberbeck-Boussinesq equations [2,3], and their expressions can be given as follows [6,8].
(5)$$\frac{\partial \rho }{\partial t}+\nabla \cdot (\rho u)=0$$
(6)$$\frac{\partial (\rho u)}{\partial t}+u\cdot \nabla (\rho u)=-\nabla p+\nabla \cdot (2\rho \nu S)-g\beta \Delta \theta$$
(7)$$\frac{\partial \theta }{\partial t}+u\cdot \nabla \theta =\kappa \nabla^{2}\theta$$
in which ρ is the fluid density, u represents the macroscopic velocity, ν is the kinematic viscosity, p is the pressure of fluid, κ denotes the diffusivity, β is the thermal diffusion coefficient, g is the force of gravity, Δθ represents the difference of temperature and θ denotes macroscopic temperature of fluid.

A large number of numerical methods are widely used to solve the classical Oberbeck-Boussinesq equations [31,32,33,34,35,36]. The finite element methods [36], finite difference method [34] and the finite volume method [35] are traditional macroscopic methods for Computational Fluid Dynamics (CFD) calculation. The lattice Boltzmann method (LBM) is a computational fluid dynamics method based on mesoscopic simulation scale [37,38,39,40,41]. Compared with other traditional CFD calculation methods, this method has mesoscopic model characteristics between a micromolecular dynamics model and a macrocontinuous model. LBM also has the advantage of a simple description of fluid interaction, and is easier to set a complex boundary, reach a parallel calculation, and implement a program, etc. [42]. LBM has been widely considered as an effective method to describe fluid motion and deal with engineering problems [43,44]. In the subsection, double distribution LBM for will be introduced.

### 2.2. Numerical Method for Convection Diffusion Equation of Thermal Fluid

In the present paper, the double distributions of LBM are implemented to study the convection diffusion equation of thermal fluid, respectively [38,39]. A lattice Boltzmann equation is implemented to simulate the fluid flow field. Its expression is as follows [42]:(8)$$f_{i}(x+c_{i}\Delta t,t+\Delta t)=f_{i}(x,t)+(f_{i}^{eq}(x,t))-f_{i}(x,t))/\tau_{\nu }+F_{i}$$
where $f_{i}(x,t)$ denotes the density distribution functions at (x, t), $c_{i}$ represents the discrete velocity. $F_{i}$ is the discrete force term, $f_{i}^{eq}(x,t)$ is the equilibrium function of density distribution, and $\tau_{\nu }$ denotes the relaxation time. The equilibrium function for the density is given as:(9)$$f_{i}^{eq}=\rho w_{i}[1+\frac{c_{i}\cdot u}{c_{s}^{2}}+\frac{{\left(c_{i}\cdot u\right)}^{2}}{c_{s}^{2}}-\frac{u^{2}}{2c_{s}^{2}}]$$
where $w_{i}$ denotes the weight coefficient [42]. The kinematic viscosity ν is computed by the following equation
(10)$$\nu =\frac{2\tau_{\nu }-1}{6}\frac{{\left(\Delta x\right)}^{2}}{\Delta t}$$

The lattice Boltzmann equation for the temperature field is given by the following equation
(11)$$g_{i}(x+c_{i}\Delta t,t+\Delta t)=g_{i}(x,t)+(g_{i}^{eq}(x,t))-g_{i}(x,t))/\tau_{\theta }$$
where $g_{i}(x,t)$ is the temperature distribution function at (x, t), $\tau_{\theta }$ denotes the relaxation times for temperature evolution equation. $g_{i}^{eq}$ is the equilibrium function for temperature distribution. Its expression is given as [40]:(12)$$g_{i}^{eq}=\theta w_{i}[1+\frac{c_{i}\cdot u}{c_{s}^{2}}+\frac{{\left(c_{i}\cdot u\right)}^{2}}{c_{s}^{2}}-\frac{u^{2}}{2c_{s}^{2}}]$$

The diffusivity κ is as follows: (13)$$\kappa =\frac{2\tau_{\theta }-1}{6}\frac{{\left(\Delta x\right)}^{2}}{\Delta t}$$

The density, macroscopic velocity, and temperature are as follows: (14)$$\rho =\sum_{i=0}^{8}f_{i},\;\;\rho u=\sum_{i=0}^{8}c_{i}f_{i},\;\theta =\sum_{i=0}^{8}g_{i}$$

The Chapman–Enskog expansions of Equations (8) and (11) are used to derive the classical Oberbeck-Boussinesq equations [42]. A macroscopic length scale (*x*_1_ = *εx*) and two macroscopic time scales (*t*_1_ = *εt*, *t*_2_ = *εt*) are implemented in the Chapman–Enskog expansion. Two time scales ∂*t* = *ε*∂*_t_*_1_ + *ε*^2^∂*_t_*_2_ and one spatial scale ∂*_x_* = *ε*∂*_α_* are used for the Frisch, Hasslacher, and Pomeau (FHP) model [38]. The classical Oberbeck-Boussinesq equations can be derived by executing the streaming step and using the above Chapman-Enskog expansion [42].

Figure 1 shows the computational model of geometrical schematic. As shown in Figure 1, the inhomogeneities heat plates are restricted only in the bottom condition (*y* = 0), and are made of alternating regions of either the isothermal boundary condition, *θ* = *θ*_down_, where the discrete black region denotes heat source, or adiabatic boundary condition, ∂*_y_θ* = 0. The upper boundary keeps a constant temperature, *θ* = *θ*_up_. In this physical model, the ratio of width of dividing height, *ξ* = *H*/*L*, and another two nondimensionless parameters are implemented to define the geometrical configuration of the discrete heat source; the ratio of single heat source is defined as *λ* = l/*L*, and the total ratio of discrete heat source area, *η* = *nl*/*L*, in which *n* denotes the heat source number and *l* represents the single heat-source length. When *η* is equal to 1, the model becomes the classical RB convection. The above several nondimensionless parameters are implemented to obtain a better understanding of heat transfer transport in turbulent Rayleigh-Bénard convection with mixed insulating and conducting boundary conditions.

The *Rayleigh* number is one of the most important dimensionless parameter in RB convection. Its expression is as follows: (15)$$Ra=\beta \Delta \theta gH^{3}/\nu \kappa$$

The *Nusselt* number is one of the most important dimensionless parameter in RB convection to reflect the performance of heat transfer system. It is obtained by: (16)$$Nu=1+\langle u_{y}\theta \rangle /\kappa \Delta \theta H$$
where Δθ is the temperature difference between the top boundary and the bottom boundary, H denotes the channel height, $u_{y}$ is the y-velocity, and 〈·〉 represents the average value of entire domain.

The boundary conditions play a key role in the computational stability and accuracy. The periodic boundary condition and approach of nonequilibrium extrapolation are carried out in this paper. Their ideas will be introduced, respectively. The idea of the periodic boundary condition approach is as follows [42]:(17)$$f_{i}(x,t)=f_{i}(x+L,t)$$
(18)$$g_{i}(x,t)=g_{i}(x+L,t)$$
in which the vector **L** represents length of the flow pattern. The approach of nonequilibrium extrapolation is as follows [42]:(19)$$f_{i}(x_{b},t)=f_{i}^{eq}(\rho_{w},u_{w})+(f_{i}(x_{f},t)-f_{i}^{eq}(\rho_{f},u_{f}))$$
(20)$$g_{i}(x_{b},t)=g_{i}^{eq}(\rho_{w},u_{w})+(g_{i}(x_{f},t)-g_{i}^{eq}(\rho_{f},u_{f}))$$
in which the nonequilibrium contribution can be derived from the fluid node $x_{f}$ next to $x_{b}$ along the boundary normal vector. During propagation, the unknown incoming populations can be obtained by leaving the domain at the opposite side.

As illustrated in Figure 1, the inhomogeneities heat plates are implemented in all numerical simulations. For *Ra* = 10^9^, 4000 × 2000 lattices in two-dimensional space are implemented to study the temperature fields, viscous, thermal and total entropy generation rates. The parameter λ is equal to 1/9, η is equal to 5/9, the nonequilibrium extrapolation is applied at the top and bottom boundary conditions, the periodic boundary condition is used at left and right boundaries, and the *Prandtl* number (*Pr* = *ѵ*/*κ*) is equal to 1. The dimensionless temperature of discrete heat source equals to 300 in Figure 1, and the dimensionless initial temperature of the fluid is 299. 

## 3. Results and Discussions

The analysis of temperature field, flow streamlines, and various entropy generation rates will be discussed with spatial-temporal evolution in this section, respectively.

### 3.1. Analysis of Flow and Temperature Field

Figure 2 describes the isotherms’ temperature distributions with time evolution at *t*/τ = 8, *t*/τ = 16, *t*/τ = 32, and *t*/τ = 64. Here, τ ($\tau =\sqrt{H/\beta g\Delta \theta }$) is the characteristic time of the computing system. As described in Figure 2, it can be seen that a few thermal plumes ascend in the region of dimensionless bottom boundary (0.5), two big thermal plumes descend in the region of dimensionless top boundary (1.5) at time *t*/τ = 8, and the large-scale thermal plumes descend in the region of dimensionless top boundary (0.5) and ascend in the region of dimensionless bottom boundary (1.5) at time *t*/τ = 16. According to the development phenomenon of thermal plumes at times *t*/τ = 8 and *t*/τ = 16, the thermal convective motion of the whole field is still in the initial stage of turbulent development. It was seen that with time evolution, a large-scale thermal plume ascends, strikes on the top plate, and in-volutes several thermal plumes to both sides in the left half of the system, and two large-scale thermal plumes descend at *t*/τ = 32. These thermal plumes interact and strike on the top and bottom plates with time evolution, a number of small-scale thermal plumes appear at *t*/τ = 32, which demonstrates that the physical system of thermal convection gradually evolves from the large-scale to small-scale thermal plumes with time evolution [3]. In the process of energy cascade of turbulent thermal convection, the energy of the first large vortex comes from the thermal buoyancy of the outside world, which produces the second small vortex. After the small vortex loses its stability, it produces a smaller vortex process [5]. At *t*/τ = 64, some smallest plumes can be coagulated into the big plumes, several big plumes reappear with time evolution again. The above phenomenon of temperature distributions with time evolution is consistent with the previous studies [11].

To further demonstrate the above thermal convection flow phenomenon of the whole field, the streamlines of the thermal convection flow at four same time evolution steps are shown in Figure 3. As illustrated in Figure 3, one can clearly see that two large vortexes occur in the central region of the whole field due to the injection of energy at the early characteristic time; two small vortexes appear at dimensionless bottom boundary (0.5) and dimensionless top boundary (1.5) at time *t*/τ = 8 respectively. In addition, two small vortexes ascend in the region of the dimensionless bottom boundary (0.5), two small vortexes descend in the region of dimensionless top boundary (1.5). At time *t*/τ = 16, two large vortexes in central region of the whole field become unstable, and more small vortexes appear in the dimensionless top and bottom boundaries (0.5 and 1.5). It was seen that at time *t*/τ = 32, the early large vortexes evolve into a large number of small scale vortexes, and a large number of small scale vortexes generate due to energy transfer process in the whole field. However, many small vortexes disappear in main flow field, and two relatively big vortexes reappear *t*/τ = 64. The above phenomenon demonstrates that several large vortexes interact and develop to a large number of small vortexes and a few small vortexes dissipate and big vortexes reappear with temporal evolution, which qualitatively depicts that the state of flow gradually develops from laminar flow to full turbulent thermal convection motion, and further evolve from full turbulent thermal convection to dissipation flow in the process of turbulent energy transfer.

### 3.2. Analysis of Entropy Generation Rate

The isotherms temperature distributions and streamlines with time evolution are represented in the above section and several analyses of the entropy generation rate will be discussed in the following section. Figure 4 describes the viscous entropy generation rate at times *t*/τ = 8, *t*/τ = 16, *t*/τ = 32, and *t*/τ = 64. As shown in Figure 4, it is clearly seen that the high viscous entropy generation rate mainly appears in the intersectional region between the main flow and the top and bottom boundaries and in the intersectional region between big vortexes at times *t*/τ = 8 and *t*/τ = 16. It was seen that the high viscous entropy generation rate mainly appears in the high shear region between main flow region and vortex, the low viscous entropy generation rate occurs near the central region of various vortex, which indicates that the viscous flow loss mainly occurs in the high shear region. Meanwhile, at time step *t*/τ = 32, the viscous entropy generation rate evidently increases with temporal evolution. Nevertheless, the viscous entropy generation rate evidently decreases at time *t*/τ = 64 compared to that of *t*/τ = 32, which indicates that the whole mainstream field has already entered the state of turbulent dissipation.

Figure 5 illustrates the thermal entropy generation rate at times *t*/τ = 8, *t*/τ = 16, *t*/τ = 32, and *t*/τ = 64. Plotted in Figure 5, it is obviously observed that at times *t*/τ = 8 and *t*/τ = 16, the high distribution value of thermal entropy generation rate mainly dominates in the high gradient fields of temperature, especially near the top and bottom boundaries compared with the corresponding temperature fields in Figure 2. The low distribution value of thermal entropy generation rate mainly occurs in the homogenetic temperature fields. It is seen that with spatial-temporal evolution, the high distribution value of thermal entropy generation rate gradually increases due to the interaction and strike of these thermal plumes at time *t*/τ = 32, which indicates that the order degree of thermal movement gradually tends to be disordered in the whole closed system. However, the plume scale of thermal entropy generation rate gradually decreases at time *t*/τ = 64 compared to that of *t*/τ = 32, which further demonstrates that a great deal of large scale turbulent structures interact and develop into a large number of small scale turbulent structures; the thermal dissipation also appears with time evolution in the closed system. 

Figure 6 describes the total entropy generation rate at times *t*/τ = 8, *t*/τ = 16, *t*/τ = 32, and *t*/τ = 64. As described in Figure 6, it can be seen that the high distribution value of total entropy generation rate mainly dominates in the largest temperature velocity gradient compared with the corresponding temperature fields in Figure 2. The low total entropy generation rate mainly clusters in the region of the homogenetic temperature fields. The distribution size trend of total entropy generation rate is well consistent with that of thermal entropy generation rate in the corresponding time step. In the spatial evolution, the shape of high entropy generation rate congeals into a large number of varied plumes, which indicates that the role of thermal entropy generation rate gradually improves with time evolution in the heat transfer irreversibility. It can be clearly seen that with time evolution, a great deal of large scale plumes interact and develop to a large number of small scale plumes in the closed system, and the value of total entropy generation rate increases, which indicates that the order degree of energy dissipation in the whole closed system gradually tends to be disordered and increase. The viscous, thermal and total entropy generation rates with evolution can promote the idea that the type of mixed bottom boundary condition and thermal configuration can be extensively applied in a wide variety of practical engineering applications, such as the solar thermal absorber plate or the electronic existing plates.

### 3.3. Quantitative Analysis of Entropy Generation Rate with Time Evolution

The probability density function (PDF) is used to reveal the distribution aggregation situation of physics variable. Wei et al. [24] argued that the PDFs of *S_u_*, *S_θ_* and *S* with increase of Prandtl number, the tails of high entropy generation rates can fit well into the curve of the log-normal coordinate and the departure and the distribution of log-normality, gradually becoming more robust with the decrease of Prandtl number. In this paper, an exponential expression is implemented for PDF. Its exponential expression is as follows [24]:(21)$$p(Y)=\frac{C}{\sqrt{Y}}\exp(-mY^{\alpha })$$
in which *m*, *α* and *C* represent the fitted parameters, and *Y* = *X* − *X*_mp_ with *X* = *S_u_/(S_u_)*
_rms_, *S_θ_/(S_θ_) _rms_*, *S/(S) _rms_* and *X*_mp_ being the abscissa of the most probable amplitude. The best fit of Equation (21) to the data yields *m* = 0.86 and *α* = 0.72 for *S_u_*, *m* = 1.15 and *α* = 0.69 for *S_θ_* and *m* = 1.06 and *α* = 0.72 for S.

To highlight the distribution aggregation differences of $\overset{\cdot }{S_{u}}$, $\overset{\cdot }{S_{\theta }}$ and $\dot{S}$ with time evolution, the PDFs of $\overset{\cdot }{S_{u}}$, $\overset{\cdot }{S_{\theta }}$ and $\overset{\cdot }{S_{0}}$ are plotted, respectively, where $\overset{\cdot }{S_{u}}$,$\overset{\cdot }{S_{\theta }}$ and $\overset{\cdot }{S_{0}}$ represent the value distributions of *S_u_*, *S_θ_* and *S* in the whole region. Figure 7 describes the PDFs’ distributions of $\overset{\cdot }{S_{u}}$ at four times *t*/τ = 8, *t*/τ = 16, *t*/τ = 32 and *t*/τ = 64. As described in Figure 7, we can see that the high value of $\overset{\cdot }{S_{u}}$ decreases in a range of $\overset{\cdot }{S_{u}}>10$ with time evolution. This is mainly due to the fact that the flow characteristic velocity of the large-scale flow in early characteristic time at *t*/τ= 8 is relatively large, the large-scale flow is broken into more small-scale flows, the viscosity entropy rate decreases in high value with time evolution, and the viscosity entropy generation rate of the small-scale flow is smaller than that of the large-scale flow.

Figure 8 shows the PDF distributions of thermal entropy generation rate at four times *t*/τ = 8, *t*/τ = 16, *t*/τ = 32, and *t*/τ = 64. Plotted in Figure 8, it is clearly obtained that the high values of $\overset{\cdot }{S_{\theta }}$ keeps almost the same in a wide range of $\overset{\cdot }{S_{u}}>100$ with time evolution, the low and middle values of $\overset{\cdot }{S_{\theta }}$ keep light difference in a wide range of $\overset{\cdot }{S_{u}}<100$ with time evolution. Figure 9 illustrates the PDF distributions of total entropy generation rate $\overset{\cdot }{S_{0}}$ at four times *t*/τ = 8, *t*/τ = 16, *t*/τ = 32, and *t*/τ = 64. As illustrated in Figure 9, it can be seen that the high values of $\overset{\cdot }{S_{0}}$ keep almost the same with $\overset{\cdot }{S_{\theta }}$, which indicates that the thermal entropy generation rate has a dominant position in the total entropy generation rate with time evolution.

To further reveal the distribution differences of *S_u_*, *S_θ_* and *S* with time evolution, the average value of $\overset{\cdot }{S_{u}}$, $\overset{\cdot }{S_{\theta }}$ and $\overset{\cdot }{S_{0}}$ are plotted in the whole region, respectively. $\bar{\overset{\cdot }{S_{u}}}$,$\bar{\overset{\cdot }{S_{\theta }}}$, and $\bar{\overset{\cdot }{S_{0}}}$ denote the average value of $\overset{\cdot }{S_{u}}$,$\overset{\cdot }{S_{\theta }}$ and $\overset{\cdot }{S_{0}}$ in the whole region. Figure 10 shows the time evolution of average viscous entropy generation rate from *t*/τ = 0 to *t*/τ = 100 in the whole field. Plotted in Figure 10, it is clearly observed that the average value of $\bar{\overset{\cdot }{S_{u}}}$ alternately increases, three peaks successively appear from the time step of *t*/τ = 0 to *t*/τ = 32 with time evolution. One strong peak appears at the time step of *t*/τ = 32, however, the average value of $\bar{\overset{\cdot }{S_{u}}}$ gradually decreases from the time step of *t*/τ = 32 to *t/*τ = 64, and the average value of $\bar{\overset{\cdot }{S_{u}}}$ gradually increases in a range of *t*/τ > 64. This is mainly due to the fact that the largest length-scales eddy is produced owing to the injection of energy at an early characteristic time; the decrease of flow eddies and the geometric eddy size is associated with the characteristic time scales (*t*/τ < 32). However, with time evolution, the large-scale flow is broken into more small-scale flows in a range of *t*/τ from 32 to 64; the viscosity entropy rate decreases in high value with time evolution, and the viscosity entropy generation rate of the small-scale flow is smaller than that of the large-scale flow. In a range of *t*/τ > 64, some of the smallest eddies can be distorted in this distortion process, which further indicates that the kinetic energy may be dissipated from the dissipation of the smallest eddies owing to the effect of viscous flow.

Figure 11 illustrates the time evolution of average thermal entropy generation rate from *t*/τ = 0 to *t*/τ = 100 in the whole field. As illustrated in Figure 11, one can clearly see that at first $\bar{\overset{\cdot }{S_{\theta }}}$ is very large due to the extremely thin boundary layer. As time goes by, the boundary layer thickness rapidly increases to the normal level, and $\bar{\overset{\cdot }{S_{\theta }}}$ decreases rapidly. After the initial period, the average value of the temperature generation rate $\bar{\overset{\cdot }{S_{\theta }}}$ alternately increases, several peaks periodically appear from the time step of *t*/τ = 0 to *t*/τ = 32 with time evolution. One strong peak appears at the time step of *t*/τ = 32, however, the average value of $\bar{\overset{\cdot }{S_{\theta }}}$ periodically decreases from the time step of *t*/τ = 32 to *t*/τ = 64. The average value of $\bar{\overset{\cdot }{S_{\theta }}}$ periodically and lightly increases in a range of *t*/τ > 64. This is mainly due to the fact that the large scale plumes produce owing to the injection of energy at the early characteristic time, the decrease of thermal plumes size is associated with the characteristic time-scales (*t*/τ < 32). Nevertheless, with time evolution, the large-scale plumes are broken into more small-scale plumes in a range of *t*/τ from 32 to 64, the thermal entropy rate decreases in high value with time evolution, and the thermal entropy generation rate of the small-scale plumes is smaller than that of the large-scale plumes. In a range of *t*/τ > 64, some of the smallest plumes can be coagulated into the big plumes.

Figure 12 shows the time evolution of average total entropy generation rate from *t*/τ = 0 to *t*/τ = 100 in the whole field. As shown in Figure 12, one can clearly see that at first $\bar{\overset{\cdot }{S_{0}}}$ is very large due to the extremely thin boundary layer. With time evolution, it is clearly seen that the boundary layer thickness rapidly increases to the normal level, and $\bar{\overset{\cdot }{S_{0}}}$ decreases rapidly. Plotted in Figure 12, it can be seen that the high values of $\overset{\cdot }{S_{0}}$ remain almost the same with $\overset{\cdot }{S_{\theta }}$ with time evolution, which indicates that the thermal entropy generation rate plays a dominated role in the heat transfer irreversibility—the viscous entropy generation can be neglected time evolution. The above phenomenon is well consistent with the importance of heat transfer irreversibility in the previous studies [23,24,25]. Wei et al. [24] studied the effect of changing the Prandtl number on the entropy generation rate in two-dimensional RB convection, and argued that the thermal entropy generation rate has a dominant role in the heat transfer irreversibility—the viscous entropy generation can be neglected with the increasing Prandtl number. Mohamed et al. [45,46,47] studied a new analytical solution of a longitudinal fin with variable heat generation and thermal conductivity in the mixed convection Falkner-Skan flow of nanofluids with variable thermal conductivity.

## 4. Conclusions

In this paper, the time evolution features of entropy generation rate in turbulent Rayleigh-Bénard convection are investigated in mixed insulating and conducting boundary conditions. Several conclusions are given as follows.

The physical system of thermal convection gradually evolves from the large-scale to small-scale thermal plumes—some of the smallest plumes can be coagulated into the big plumes and several big plumes reappear in the time evolution. The state of flow gradually develops from laminar flow to turbulent thermal convection motion, and further evolves from turbulent thermal convection to dissipation flow in the process of turbulent energy transfer.

The viscous, thermal, and total entropy generation rates evidently increase in wide range of *t*/τ < 32 with temporal evolution. Nevertheless, the viscous, thermal, and total entropy generation rates evidently decreases at time *t*/τ = 64 compared to that of *t*/τ = 32.

The high value of $\overset{\cdot }{S_{u}}$ decreases in a range of $\overset{\cdot }{S_{u}}>10$ with time evolution. It is revealed that the flow characteristic velocity of the large-scale flow in early characteristic time at *t*/τ = 8 is relatively large: the large-scale flow is broken into more small-scale flows, the viscosity entropy rate decreases in high value with time evolution, and the viscosity entropy generation rate of the small-scale flow is smaller than that of the large-scale flow.

It was seen that the largest length-scale eddy produces owing to the injection of energy at the early characteristic time, the decrease of flow eddies and geometric eddy size are associated with the characteristic time-scales (*t*/τ < 32). However, the large-scale flow is broken into more small-scale flows in a range of *t*/τ from 32 to 64, the viscosity entropy rate decreases in high value with time evolution, and the viscosity entropy generation rate of the small-scale flow is smaller than that of the large-scale flow. In a range of *t*/τ > 64, some of the smallest eddies can be distorted in this distortion process. The average value of $\bar{\overset{\cdot }{S_{\theta }}}$ alternately increases from the time step of *t*/τ = 0 to *t*/τ = 32, however, the average value of $\bar{\overset{\cdot }{S_{\theta }}}$ periodically decreases from the time step of *t*/τ = 32 to *t*/τ = 64. Interestingly, the average value of $\bar{\overset{\cdot }{S_{\theta }}}$ periodically and lightly increases in a range of *t*/τ > 64. 

The above studies further demonstrate the process of the energy injection, the kinetic energy, the kinetic energy transfer, the thermal energy transfer, the viscous dissipated flow and thermal dissipation. In practical engineering, the type of mixed-bottom boundary condition and thermal configuration can be extensively applied in a wide variety of equipment, such as the solar thermal absorber plate or the electronic existing plates.

## Figures and Tables

**Figure 1 entropy-22-00672-f001:**
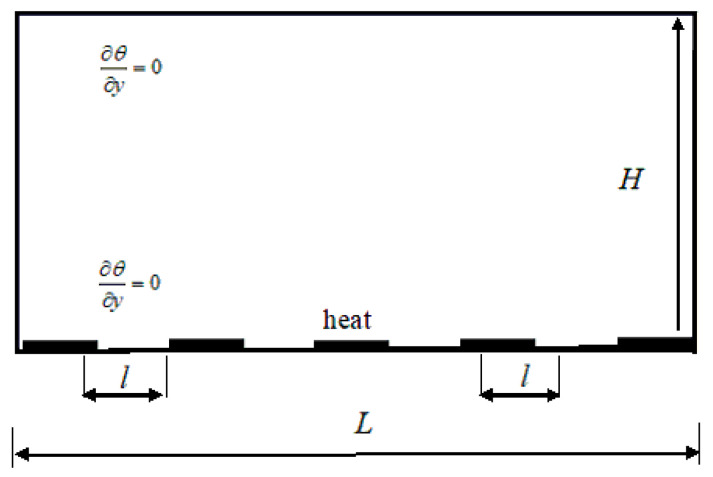
Computational geometry and boundary conditions in the two-dimensional space.

**Figure 2 entropy-22-00672-f002:**
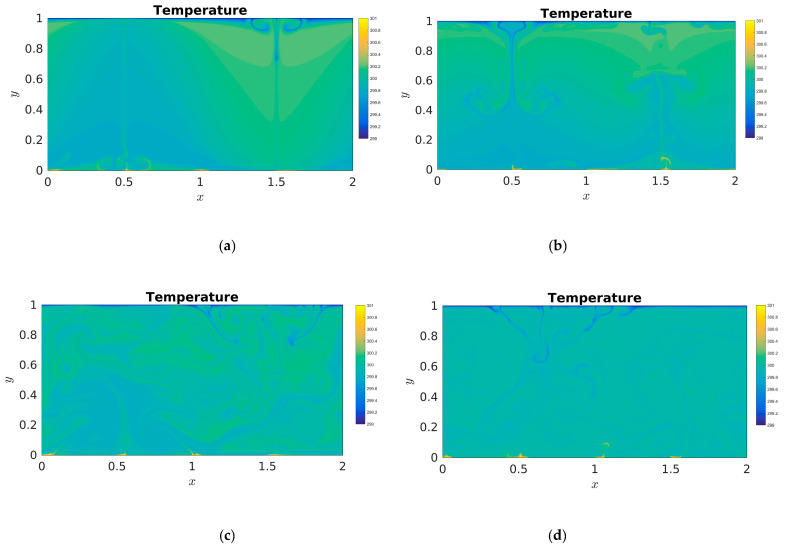
Temperature distributions (isotherms) with time evolution in the two-dimensional RB convection (**a**) *t*/τ = 8 (**b**) *t*/τ = 16, (**c**) *t*/τ = 32, and (**d**) *t*/τ = 64.

**Figure 3 entropy-22-00672-f003:**
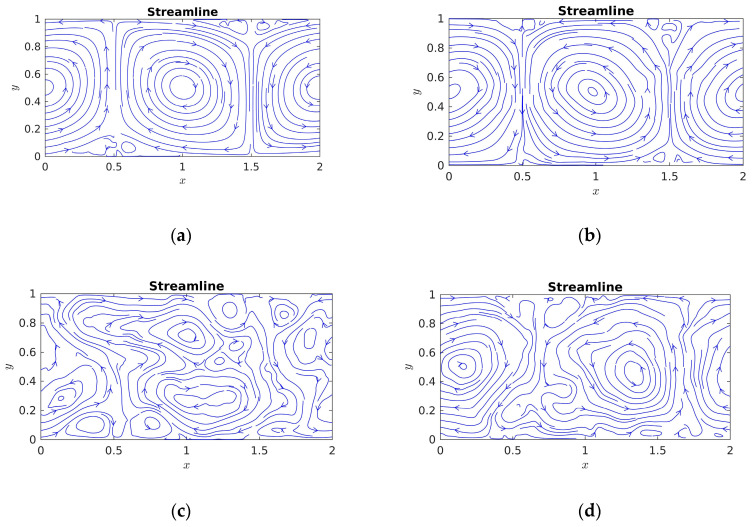
Streamlines of the thermal convection flow with time evolution in the two-dimensional RB convection (**a**) *t*/τ = 8 (**b**) *t*/τ = 16, (**c**) *t*/τ = 32, and (**d**) *t*/τ = 64.

**Figure 4 entropy-22-00672-f004:**
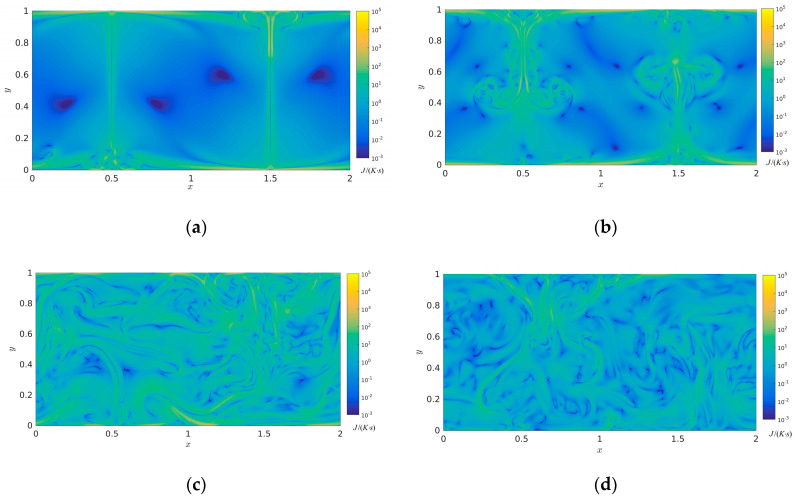
Viscous entropy generation rate with time evolution in the two-dimensional RB convection (**a**) *t*/τ = 8 (**b**) *t*/τ = 16, (**c**) *t*/τ = 32, and (**d**) *t*/τ = 64 (Units: J/(K·s)).

**Figure 5 entropy-22-00672-f005:**
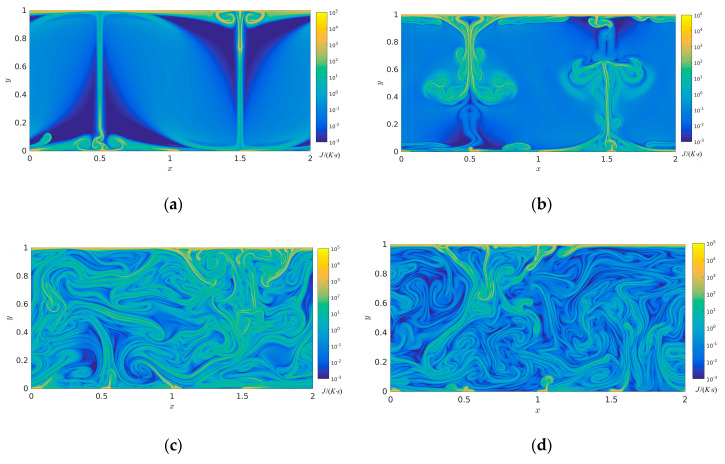
Thermal entropy generation rate with time evolution in the two-dimensional RB convection (**a**) *t*/τ = 8 (**b**) *t*/τ = 16, (**c**) *t*/τ = 32, and (**d**) *t*/τ = 64 (Units: J/(K·s)).

**Figure 6 entropy-22-00672-f006:**
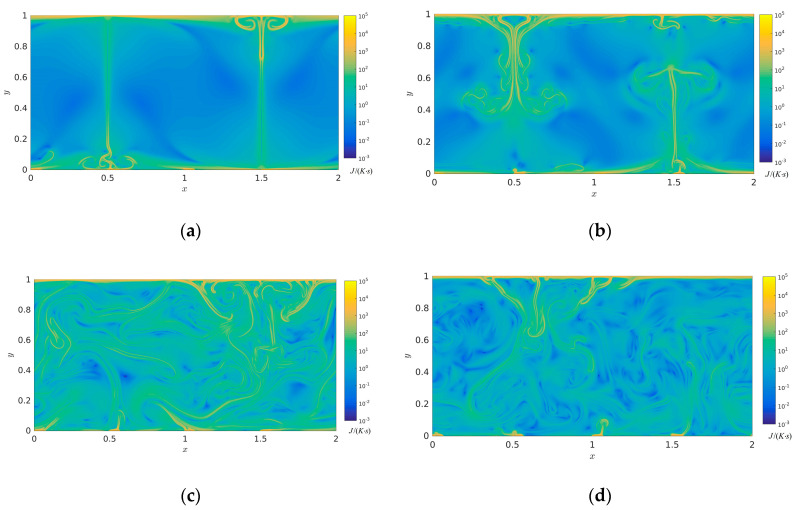
Total entropy generation rate with time evolution in the two-dimensional RB convection (**a**) *t*/τ = 8 (**b**) *t*/τ = 16, (**c**) *t*/τ = 32, and (**d**) *t*/τ = 64 (Units: J/(K·s)).

**Figure 7 entropy-22-00672-f007:**
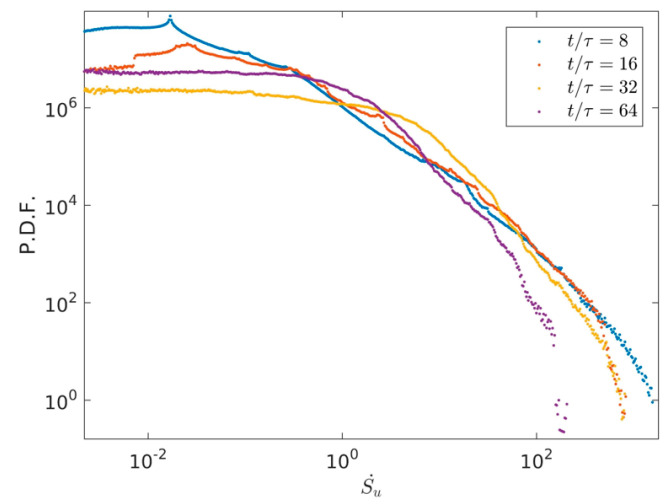
PDF distributions of viscous entropy generation rate in the two-dimensional RB convection and at four times *t*/τ = 8, *t*/τ = 16, *t*/τ = 32, and *t*/τ = 64.

**Figure 8 entropy-22-00672-f008:**
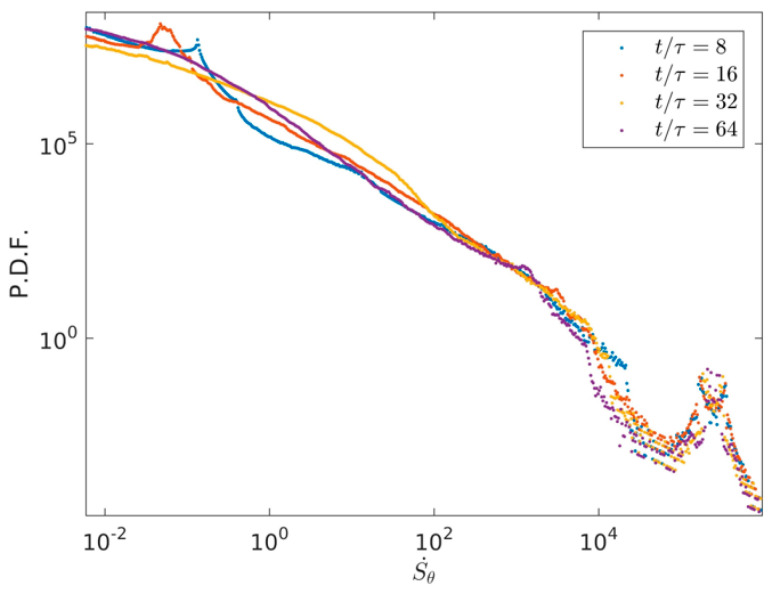
PDF distributions of thermal entropy generation rate in the two-dimensional RB convection and at four times *t*/τ = 8, *t*/τ = 16, *t*/τ = 32, and *t*/τ = 64.

**Figure 9 entropy-22-00672-f009:**
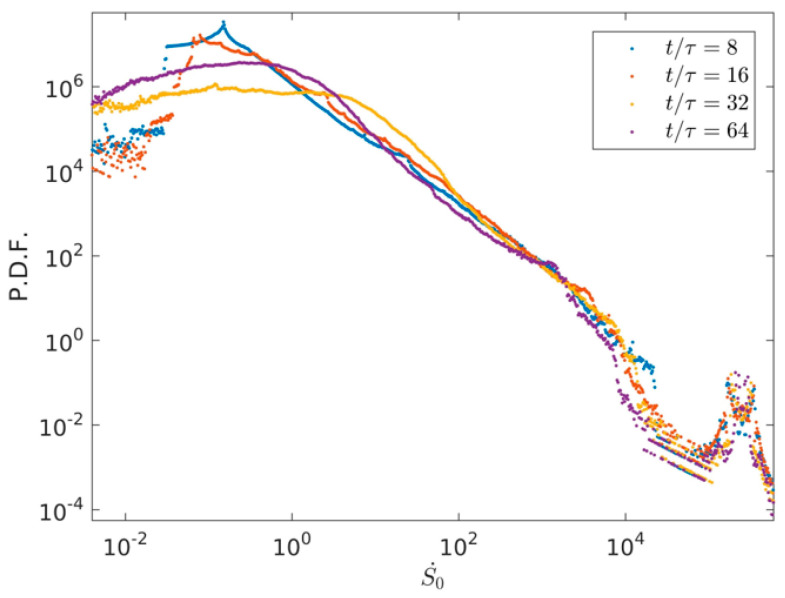
PDF distributions of total entropy generation rate in the two-dimensional RB convection and at four times *t*/τ = 8, *t*/τ = 16, *t*/τ = 32, and *t*/τ = 64.

**Figure 10 entropy-22-00672-f010:**
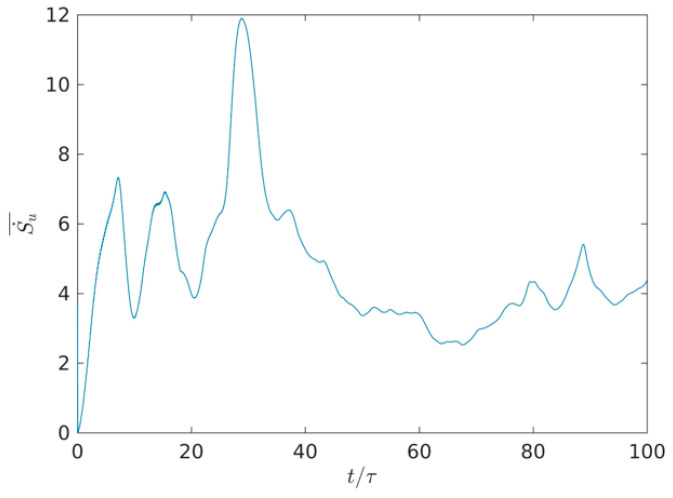
Time evolution of average viscous entropy generation rate in the whole two-dimensional field.

**Figure 11 entropy-22-00672-f011:**
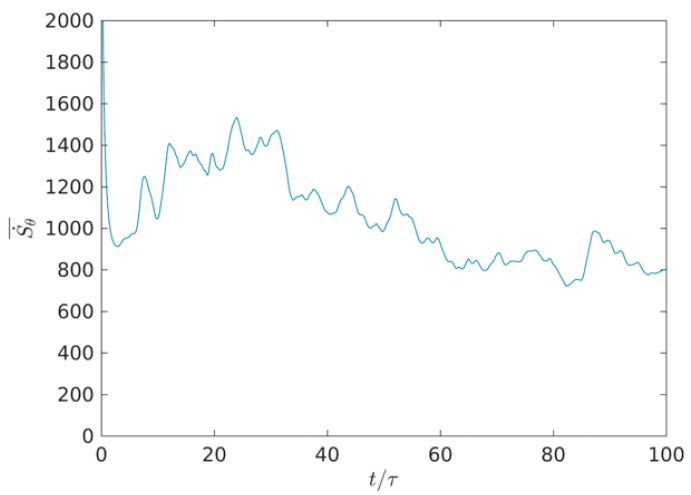
Time evolution of average thermal entropy generation rate in the whole two-dimensional field.

**Figure 12 entropy-22-00672-f012:**
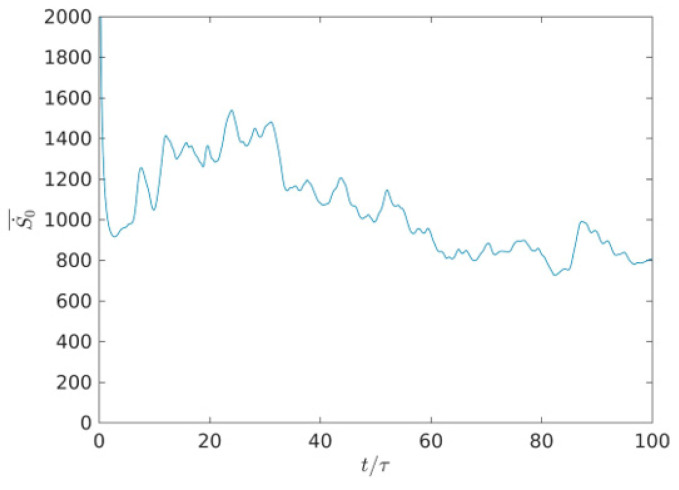
Time evolution of average total entropy generation rate in the whole two-dimensional field.

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
