# Peer review of "Time Evolution Features of Entropy Generation Rate in Turbulent Rayleigh-Bénard Convection with Mixed Insulating and Conducting Boundary Conditions"

_entropy, 2020, doi:10.3390/e22060672_

Round 1

Reviewer 1 Report

I recommend accepting the manuscript after minor revisions as suggested in the attached file. 

Author Response

Response of entropy-810394

This article is valuable and recommendable for publication. The authors should address the

following points in revisions.

  1. In the title and whole manuscript, the word “Bénard” must change to “Bénard”.

Response: Corrected.

The word “Bénard”has been replaced by the “Bénard”in the revised manuscript.

  1. The abstract can be written in a more interesting fashion and it contains some

grammatical errors such as “…evidently decreases at time…” must change to

“…evidently decrease at time…”.

Response: Corrected.

  1. The manuscript should be reviewed because it contains some punctuational mistakes and

grammatical English errors such as in the last paragraph in introduction section

“…emerge in in turbulent RB…”, “…conditions on the the time evolution…”,

“…entropy generation rates The physical insight…”, and “…discussed in in turbulent

RB…”. Certain parts of the manuscript must be revised.

Response: Corrected.

  1. Give a broader view of the literature on the topic and explain the current state-of-the-art

of the topic where the paper is framed. Considering the subject of this article, its review

should be completed by addressing the following papers:

  • The European Physical Journal Plus 135(1) (2020) 120. · International Journal for Numerical Methods in Fluids 92(1) (2020) 38-54. · Waves in Random and Complex Media (2019) 1-19.10.1080/17455030.2019.1686550
  • Journal of Non-Equilibrium Thermodynamics 45(2) (2020) 97-108.

Response: Corrected.

Four papers have been added in the revised manuscript.

  1. Some mathematical symbols are not defined.

Response: Corrected.

  1. Please go through the correctness of all mathematical equations, especially Eq. (6).

Response: Corrected.

  1. I need the clarification on the application of the problem studied.

Response: Corrected.

  1. Use of first person “We” is generally not to be done.

Response: Corrected.

  1. For validation of the problem, compare them with those available in the literature, also

including discussions on potential applications.

Response: Corrected.

  1. Why the thermal entropy generation rate plays a dominated role in the heat transfer

irreversibility. Please rewrite the discussion of Figs. 11 and 12 by including meaningful

physical interpretation for broader field audiences.

Response: Corrected.

English has been improved greatly in the revised manuscript according to reviewers’ report.

In sum, the authors appreciate sincerely the referees’ valuable comments and suggestions on this work.

[44]Chen Z. and Shu C. , Simplified  lattice  Boltzmann  method  for  non‐Newtonian  power‐law  fluid  flows,International Journal for Numerical Methods in Fluids 92(1) (2020) 38-54.

[46]Mohamed K., Ismai T, Mohamed R.E., A new analytical solution of longitudinal fin with variable heat generation and thermal conductivity using DRA,The European Physical Journal Plus 135(1) (2020) 120

[47]Nawel B, Mohamed K., Ismai T, Mohamed R.E.,On numerical and analytical solutions for mixed convection Falkner-Skan flow of nanofluids with variable thermal conductivity,Waves in Random and Complex Media (2019) 1-19.10.1080/17455030.2019.1686550

[48]Mohamed R.E.Effects of NP Shapes on Non-Newtonian Bio-Nanofluid Flow in Suction/Blowing Process with Convective Condition: Sisko Model,Journal of Non-Equilibrium Thermodynamics 45(2) (2020) 97-108.

Reviewer 2 Report

The paper should be thoroughly revised from the point of English language (syntax and grammar). I have added the pdf with some highlights in this matter.

On the other hand, there are some issues in respect of the content.

1) Eqs. 1) and 2) depict entropy production fields in dimensional form. However, all figures, starting from 4), 5) 6) and so on, lack dimensions.

2) Some important geometrical data of the RB cell are not given (H, L). If the flow is should have a Ra number of 10^9 it will result for example H.

3) What is the order relation between the temperature on the upper boundary and the temperature of the heat sources?

4) How was the entropy rate assessed in Figs. 4 and 5 to state the after an increase, a decrease of these rates is obvious?

5) The time duration of the simulation seems quite short. Over longer time, the fluctuations should diminish enough to produce sort of homogeneous fields.

6) How was the average total entropy rate computed at the initial moment? It has a very large value, and although we can consider the initial large temperature gradients at the boundaries in the first instant, from a spatial point of view these values are consistent only in a few cells? Hence, the average should be quite low.  

7) What is the relevance of the thermal boundary conditions that were imposed on the lower boundary if there is no comparison between an ordinary RB cell with uniform temperature conditions and the model the authors proposed?

8) The major conclusion is that the RB process is characteristic to turbulent energy cascade flow!!! But to reach this conclusions the simulation should take longer. 

Author Response

Response of entropy-810394

Comments and Suggestions for Authors

The paper should be thoroughly revised from the point of English language (syntax and grammar). I have added the pdf with some highlights in this matter.

On the other hand, there are some issues in respect of the content.

1) Eqs. 1) and 2) depict entropy production fields in dimensional form. However, all figures, starting from 4), 5) 6) and so on, lack dimensions.

Response: Corrected.

2) Some important geometrical data of the RB cell are not given (H, L). If the flow is should have a Ra number of 109 it will result for example H.

Response: H×L is 4000×2000(l.u.) in all simulations, the illustration is in the last paragraph in Sec. 2.

3) What is the order relation between the temperature on the upper boundary and the temperature of the heat sources?

Response: We are not quiet sure about the meaning of order relation. The non-dimensional temperature of the upper boundary is cold temperature and the non-dimensional temperature of heat sources on lower boundary is hot temperature, which are 0 and 1 in all simulations.

4) How was the entropy rate assessed in Figs. 4 and 5 to state the after an increase, a decrease of these rates is obvious?

Response: The dark blue represents low rate and light yellow represents high rate in Figs. 4 and 5. As can be seen from Figs. 4 and 5, color of contour map in subfigures (d) are lighter than that in subfigures (c). The conclusions are also can be proved by Fig. 10 and Fig.11, which clearly show the decrease in both viscous entropy generation rate and thermal entropy generation rate.

5) The time duration of the simulation seems quite short. Over longer time, the fluctuations should diminish enough to produce sort of homogeneous fields.

Response: Thanks for the valuable question. It is far too short for the statistics of turbulence. However, we mainly focus on the forming stage of the RB convection. Thus, we just focus on the first 64τ(s).

6) How was the average total entropy rate computed at the initial moment? It has a very large value, and although we can consider the initial large temperature gradients at the boundaries in the first instant, from a spatial point of view these values are consistent only in a few cells? Hence, the average should be quite low.

Response: The temperature of the fluid is initialed as (Th+Tl)/2, not only the flow field near heat source has large temperature gradient, but also the flow field near upper cold boundary has large temperature gradient either. As time goes by, the boundary layer thickness rapidly increases to the normal level, and the average temperature entropy generation rate decreases rapidly. The average temperature entropy generation rate after the initial period and before the mixture period is low, which is the mentioned situation.

7) What is the relevance of the thermal boundary conditions that were imposed on the lower boundary if there is no comparison between an ordinary RB cell with uniform temperature conditions and the model the authors proposed?

Response: We have cited the study of entropy generation rate in RB system with uniform temperature condition. In this paper, we focus on the effect of non-uniform temperature boundary condition on the forming stage of RB system.

  • The major conclusion is that the RB process is characteristic to turbulent energy cascade flow!!! But to reach this conclusions the simulation should take longer.

Response: The cascade process exists in all turbulent systems. In order to achieve a conclusion of statistical significance, it is necessary to conduct long-term statistics on fully developed turbulence. However, this phenomenological process at forming stage is discussed only by observing the flow field instead of statistics of any variables in this paper. In order to avoid misunderstanding, we have changed some of the statements of full turbulence in conclusions in the revised manuscript.

Round 2

Reviewer 2 Report

Figs. 4, 5, 6 do not show any improvement in terms of entropy generation rate dimension. The legends have no units.

Figs. 7, 8, 9, 10, 11 and 12 are missing from the pdf file.

I would suggest including a legend for Fig. 2.

Author Response

Responses of Comments

Comments and Suggestions for Authors

1. Figs. 4, 5, 6 do not show any improvement in terms of entropy generation rate dimension. The legends have no units.

Responses: Corrected.

2.I would suggest including a legend for Fig. 2.

Responses: Corrected.